# Characterization of the Gut Microbiota of Individuals at Different T2D Stages Reveals a Complex Relationship with the Host

**DOI:** 10.3390/microorganisms8010094

**Published:** 2020-01-10

**Authors:** Alejandra Chávez-Carbajal, María Luisa Pizano-Zárate, Fernando Hernández-Quiroz, Guillermo Federico Ortiz-Luna, Rosa María Morales-Hernández, Amapola De Sales-Millán, María Hernández-Trejo, Angelina García-Vite, Luis Beltrán-Lagunes, Carlos Hoyo-Vadillo, Jaime García-Mena

**Affiliations:** 1Departamento de Genética y Biología Molecular, Centro de Investigación y de Estudios Avanzados del Instituto Politécnico Nacional, Ciudad de México 07360, Mexico; alejandra.chavez@cinvestav.mx (A.C.-C.); fernando.hernandez@cinvestav.mx (F.H.-Q.); dsmn.amapola@gmail.com (A.D.S.-M.); 2Departamento de Nutrición y Bioprogramación, Instituto Nacional de Perinatología, Ciudad de México 11000, Mexico; rmh080868@yahoo.com; 3Departamento de Peri y Posmenopausia, Instituto Nacional de Perinatología, Ciudad de México 11000, Mexico; dr.ortiz.luna@gmail.com; 4Departamento de Neurolobiología del Desarrollo, Instituto Nacional de Perinatología, Ciudad de México 1100, Mexico; maria.h.trejo72@gmail.com; 5Módulo Integral de Atención al Paciente con Diabetes por Etapas, Clínica de Medicina Familiar “Gustavo A. Madero”, I.S.S.S.T.E., Ciudad de México 07020, Mexico; angyvite@gmail.com; 6Jefatura de Enseñanza e Investigación, Clínica de Medicina Familiar “Gustavo A Madero”, I.S.S.S.T.E., Ciudad de México 07020, Mexico; dr.luisbeltran007@hotmail.com; 7Departamento de Farmacología, Centro de Investigación y de Estudios Avanzados del Instituto Politécnico Nacional, Ciudad de Mexico 07360, Mexico; citocromo@cinvestav.mx

**Keywords:** gut microbiota, type 2 diabetes, prediabetes, metabolic disorders, diet, and high-throughput DNA sequencing

## Abstract

In this work, we studied 217 Mexican subjects divided into six groups with different stages of glucose intolerance: 76 Controls (CO), 54 prediabetes (PRE), 14 T2D no medication (T2D−No−M), 14 T2D with Metformin (T2D−M), 22 T2D with polypharmacy (T2D−P), and 37 T2D with polypharmacy and insulin (T2D−P+I). We aimed to determine differences in the gut microbiota diversity for each condition. At the phylum level, we found that Firmicutes and Bacteroidetes outline major changes in the gut microbiota. The gut bacterial richness and diversity of individuals in the T2D−No−M group were lesser than other groups. Interestingly, we found a significant difference in the beta diversity of the gut microbiota among all groups. Higher abundance was found for *Comamonadaceae* in PRE, and *Sutterella* spp. in T2D−No−M. In addition, we found associations of specific microbial taxa with clinical parameters. Finally, we report predicted metabolic pathways of gut microbiota linked to T2D−M and PRE conditions. Collectively, these results indicate that each group has specific predicted metabolic characteristics and gut bacteria populations for each phenotype. The results of this study could be used to define strategies to modulate gut microbiota through noninvasive treatments, such as dietary intervention, probiotics or prebiotics, and to improve glucose tolerance of individuals with prediabetes or T2D.

## 1. Introduction

Type 2 diabetes (T2D) is a complex multifactorial metabolic disorder involving genetic, environmental lifestyle factors; it is mostly characterized by glucose intolerance. T2D is major contributor to serious complications like cardiovascular disease, kidney failure, and/or lower limb amputation, among others, with a negative impact on quality of life [1]. T2D represented the eighth leading cause of global death by year in 2016 [2]. In Mexico, T2D ranks among the top ten diseases according to the Mexican National Statistical Institute (Instituto Nacional de Estadística y Geografía). In Mexico, T2D was the leading cause of death among 45 to 64, sixth among 35 to 44, and eighth among 25 to 34 year-old adults [3].

The gut microbiota is described as a complex and abundant ecosystem, composed of trillions of microorganisms [4]. In adults, most bacteria usually belong to the abundant Bacteroidetes and Firmicutes bacterial phyla, while others like Actinobacteria, Fusobacteria, and Verrucomicrobia vary among individuals [5].

The gut microbiota has been proposed as an essential environmental factor involved in energy metabolism, and as being associated with metabolic disorders such as obesity and T2D [6]. There is evidence suggesting that gut microbiota plays an important role in T2D pathogenesis [7]. This pathogenesis involves different mechanisms such as metabolic endotoxemia, dysfunction of the LPS/CD14/TLR4 and endocannabinoid systems, altered mucosa permeability, reduction of intestinal alkaline phosphatase activity, defective fermentation of dietary polysaccharides, and increased intestinal absorption of monosaccharides and lipid metabolism [8].

There are other mechanisms suggesting a possible bacterial translocation from the gut to the bloodstream associated with T2D. In the Japanese population, it has been reported that, in addition to dysbiosis of the gut microbiota, there is evidence of live gram-positive bacteria in the peripheral bloodstream, detected more abundantly in people with T2D (28%) than in healthy subjects (4%) [9].

Some studies have monitored the gut microbiota and its relationship with T2D in different populations. One study in China with 345 individuals determined that people with T2D had a moderate bacterial dysbiosis with respect to the control group, with a statistically significant increase in *Clostridium bolteae*, *C. hathewayi*, and the family *Lachnospiraceae* [10]. In another study, with 145 European women over 70 years-old with glucose intolerance, it was determined that the observed changes in the gut microbiota including increases in *Clostridium clostridioforme* and decreases in the abundance of *Roseburia* could be directly related to the development of T2D [11]. Moreover, in a study of 63 Mexican-Americans (76% women) with high risk of T2D and obesity, differences in Firmicutes and Bacteroidetes abundances were observed in comparison to the control group; these changes were the major contributing factors for the development of the disease [12]. Also, a study of 38 Estonians (60% women) suggested that hyperglycemia can be predicted by a reduction in the abundance of some gut anaerobic bacteria, like *Bacteroides* spp. [13].

It has been shown that oral pharmacological therapy with Metformin, the first line monotherapy drug to control T2D, alters the gut microbiota composition, with observed changes after two months of treatment. In this report, an increase in the number of positive correlations among bacterial genera, especially those in the Proteobacteria and Firmicutes phyla, was observed [14]. With this evidence, it appears plausible to identify profiles of gut microbiota related to the developmental stages of this disease, and that occur prior to the appearance of complications such as hypertension and hyperlipidemias.

The purpose of our work was to characterize the gut microbiota in individuals at different stages of T2D development, with and without pharmacological treatment. We aimed to obtain an insight into the diversity profile and type of bacteria found at each stage of this disease. We believe that T2D treatments could profit from this knowledge, for the design of novel therapies such as gut microbiota modulation with dietary interventions, and the use of probiotics, prebiotics, and/or fecal microbiota transplantations that ultimately improve glucose metabolism and insulin resistance in patients.

## 2. Materials and Methods

### 2.1. Study Subjects

The study design was cross-sectional, aiming to characterize the gut microbiota in members of the studied groups. All the study subjects were recruited in the hospitals where they received medical attention for their condition in Mexico City (“Clínica de Medicina familiar Gustavo A. Madero” and the “Instituto Nacional de Perinatología”). The cases (T2D-M, T2D-P and T2D-P+I) were derived from the “Comprehensive Program of Patient Care with Diabetes by Stages”, in which the medical treatment was verified by the medical staff. The same was true for the controls (CO), prediabetes (PRE) and type 2 diabetes no medication (T2D-No-M), recruited at the hospital while escorting a family member. After agreeing to participate, all individuals were classified accordingly into groups using the American Diabetes Association (ADA) criteria. Participants in the groups were classified based on the HbA1c% using the cut-off points: CO ≤ 5.6%, PRE 5.7–6.4%, and T2D ≥ 6.5%. The cut-off points for fasting glucose levels were CO < 100 mg/dL, PRE 100 to < 126 mg/dL, and T2D ≥ 126 mg/dL. The HbA1c% agreed with the fasting glucose level for each participant. The PRE and T2D-No-M individuals were immediately assigned to the medical care unit for treatment after samples were collected for the study. The recruitment process occurred from 11 November 2015 to 11 October 2016. The studies included 217 Mexican subjects (143 women and 74 men) that satisfied the ADA criteria, with an average age of 49 years old. The inclusion criteria were people who decided to participate in the study and provided a blood sample to obtain plasma and feces for study. In the case of T2D patients, those who were diagnosed by parameters established by the ADA and who were between 40 and 55 years old were included. The exclusion criteria were gastrointestinal diseases, antibiotic treatment, and prebiotics or probiotics intake during the last three months, vegetarians and vegans, allergies, and autoimmune disease. The participants were divided into six groups: 76 in Control (CO); 54 in Prediabetes (PRE); 14 in T2D with no medication (T2D−No−M); 14 in T2D with Metformin medication (T2D−M) for one to two years; 22 in T2D having an average of seven years of pharmacological treatment for the control of other diseases, such as hyperlipidemia (bezafibrate and/or pravastatin), and/or hypertension (amlodipine and/or losartan) (T2D−P); finally, 37 in a group with same characteristics as T2D−P, but with insulin as part of the pharmacological polypharmacy treatment (T2D−P+I). All participants provided informed consent in accordance with the Helsinki Declaration revised in 2013. The ethics and scientific committees from the “Clínica de Medicina Familiar Gustavo A. Madero, project identification code 20151030, 29 April 2016” and “Instituto Nacional de Perinatología project identification code 212250-3310-21108-04-1, 30 May 2016” approved the research protocols.

### 2.2. Anthropometrical Evaluation

The percentage of fat mass and body weight in kilograms of the participants was measured using an InBody Model 720 (± 0.1 kg accuracy, 250 kg capacity, Biospace Co., Seoul, Korea), under fasting conditions defined as no caloric or water intake for at least 8 h. The subjects stood in the InBody with minimum clothing, placing the soles and palms on the electrodes, to measure the fat and the fat-free mass percentages [15]. All percentages of body fat and fat-free mass were confirmed by the air displacement plethysmography method (Bod Pod) [16]. Height was measured in a standing position after removing shoes using a stadiometer (Seca^®^ Model 240; ±2 mm accuracy, Seca GmbH & Co., Hamburg, Germany). The body mass index (BMI) was calculated as the weight in kilograms divided by the squared height in meters. The BMI from 18.50 to 24.99 kg/m^2^ was categorized as normal; from 25.00 to 29.99 kg/m^2^ as overweight; and >30.00 kg/m^2^ as obese. The waist circumference was measured at the midpoint between the lower rib and iliac crest. The hip circumference was measured under the iliac crest using a metric tape, in all subjects (Lufkin, model W606ME, Lufkin Board, Cleveland, OH, USA).

### 2.3. Dietary Assessment

A registered dietitian performed a “24-h Dietary Recall” to assess detailed information about all food and beverages consumed in one day. We calculated the daily intake of total kilocalories, macronutrients, and sugar with the use of data about nutrients, measures, and other related information from the FoodData Central, U.S. Department of Agriculture (https://fdc.nal.usda.gov/) and the Mexican Food Equivalent Composition List “Sistema Mexicano de Equivalentes” [17].

### 2.4. Types of Physical Activity

The “Global Physical Activity Questionnaire” (GPAQ), validated by the World Health Organization [18], was applied to classify the types of physical activity. The results were reported as Metabolic Equivalents (METs) in hours and minutes, including those that resulted from activities at work, transportation, recreational, and sedentary behavior. The units of the METs for each type of activity of the GPAQ questionnaire were obtained from the “Compendium of Physical Activities from the American College of Sports Medicine” [19]. Once the METs were determined for all the GPAQ activities, the average METs per day were calculated. Subsequently, the resulting METs were classified according to the type of physical activity proposed by the “American College of Sports Medicine” [19] as follows: sedentary behavior 1.0–1.5 METs, light intensity 1.6–2.9, moderate intensity 3.0–5.9 METs, vigorous to intense ≥ 6 METs.

### 2.5. Biochemical Profile

During the study, all the serum and plasma were immediately stored at −70 °C after collection. Fasting glucose was determined by the glucose oxidase reaction, measured by the photometric method. Total cholesterol, triglycerides, HDL-cholesterol, and LDL-cholesterol levels were determined by enzymatic colorimetric tests. To determine the concentration of HbA1c, venous blood was collected in heparinized blood collection tubes (BD, Vacutainer, Franklin Lakes, NJ, USA) and then used in immunoturbidimetric tests with reinforcement particles (Innovastar, Diasys Diagnostic Systems, Holzheim, Germany). Tests were standardized based on the approved reference method from the International Federation of Clinical Chemistry (IFCC).

### 2.6. DNA Extraction from Feces

Fecal samples were obtained from the participants at the hospital and immediately stored at −70 °C until DNA was extracted. The DNA was extracted as previously described [20] except for using PowerSoil^®^ DNA Isolation Kit (Cat# 12888-50, MO BIO Laboratories Inc., Carlsbad, CA, USA) which utilizes patented Inhibitor Removal Technology^®^ (IRT) beads for cells lysis, DNA binding, washing, and elution chromatographic purification steps. Quantity of purified coproDNA was measured at 260/280 absorbance using a NanoDrop Lite Spectrophotometer (Thermo Scientific, Waltham, MA, USA). The quality was evaluated by electrophoretic fractionation in 0.5% agarose gels.

### 2.7. Construction of the V3-16S rRNA Gene Library and High Throughput DNA Sequencing

For each participant, an amplicon of approximately 281 bp containing the V3 variable region of the 16S rDNA gene was amplified using V3‒341F forward primer (set of barcodes) complementary to positions 340–356 of the *Escherichia coli* 16S rDNA gene molecule *rrnB* GenBank J01859.1, and the V3‒518R reverse primer complementary to positions 517–533. The PCR mixture was 1× PCR buffer (5 mM KCl, 1 mM Tris-HCl pH 8.0), 2 mM MgCl_2_, 0.2 mM of dNTPs (Cat. R0193 Thermo Scientific), 0.2 μM 16S rRNA forward barcoded and reverse primers, 0.05 U of Phusion High-Fidelity DNA Polymerase (Cat. F-530L Finnzymes-Thermo Scientific), and 20 ng of template DNA in a final volume of 50 μL. The thermocycler programming was: 5 min at 95 °C; 25 cycles of 15 s at 94 °C, 15 s at 62 °C and 15 s at 72 °C; subsequently 10 min at 72 °C. The amplification of the DNA of each sample was performed by PCR GeneAmp System 2700 Thermocycler (Applied Biosystems, Beverly, MA, USA). PGM sequencing was made in-house using Ion OneTouch 2, Ion PGM Template OT2 200 Kit v2 DL (Life Technologies, Camarillo, CA, USA), Ion 318 Chip Kit v2 and Ion Torrent PGM System (Guilford, CT, USA). After sequencing, reads were filtered by the PGM software to exclude low quality and polyclonal sequences obtaining reads with Phred33 Quality Score of 31 on average. All reads were trimmed to 200 nt using FastQC. Filtered and demultiplexed FASTQ files were converted into FASTA files, concatenated into a single file and then processed with multiple QIIME (Quantitative Insights into Microbial Ecology) v1.9.1 scripts (http://qiime.org/scripts/) [20]. The DNA sequences were classified into 33,000 Operational Taxonomic Units (OTUs) using closed based picking parameters with a 97% similarity level against the Greengenes database v13.8. (http://qiime.org/home_static/dataFiles.html). We analyzed 217 samples obtaining a minimum of 10,907 and a maximum of 382,307 reads, with an average of 120,947 reads per sample. Sequences were deposited in NCBI BioProject repository, accession number PRJNA472187 and can be accessed through the following link: http://www.ncbi.nlm.nih.gov/bioproject/472187.

### 2.8. Microbial Diversity Analysis

Alpha diversity was generated with the script alpha_rarefaction.py in QIIME, using the metrics: Chao1, Observed Species, Shannon, and Simpson to obtain the respective indexes. To find different significances among groups we used compare_alpha_diversity.py script based on a two-sample *t*-test using the default number of Monte Carlo permutations. The graphics were generated by R v3.4.2 using the phyloseq and ggplot2 libraries [21]. The Beta microbial diversity was made using the beta_diversity.py script of QIIME. To identify significant differences between all groups, we used the compare_categories.py script of QIIME, using a distance matrix as the primary input and mapping file. As a statistical method, we used ANOSIM, a nonparametric test in which its significance was determined through 99 permutations.

### 2.9. Analysis of Bacterial Enrichment among Study Groups

Linear Discriminant Analysis Effect Size (LEfSe) was used to identify the significant enrichment of bacteria in the gut microbiota. The LEfSe analysis was calculated with the relative abundance information file of all bacteria obtained at Class, Order, Family and Genus taxonomical level. We used the script run_lefse.py to obtain *p*-values for the factorial Kruskal-Wallis test among classes, and *p*-values to obtain pairwise Wilcoxon. Only *p*-values < 0.05 were considered significant. After this step, we used the script plot_res.py to generate the figures [22]. Finally, we made a Benjamini-Hochberg correction with the p.adjust() function in R v3.4.2 to eliminate False Discovery Rate (FDR) [23,24].

### 2.10. Predictive Functional Metagenome of the Gut Microbiota using PICRUSt

The prediction analyses of the functional genes in the gut microbiota for all the studied groups were performed using PICRUSt. The taxonomy assignment was made with Greengenes database v13.8 with a 97% similarity. The OTU table was normalized with the PICRUSt workflow (http://galaxy.morganlangille.com) to obtain the final metagenome functional prediction [25]. To analyze the data, we used the Statistical Analysis of the Metagenomic Profiles (STAMP) software v2.1.3 using Welch’s *t*-test, and Benjamini-Hochberg as multiple test correction [26].

### 2.11. Multivariate Association of the Gut Microbiota with All Measured Variables

To obtain associations of gut bacteria populations with the metadata, we used a Multivariate Association with linear Models (MaAsLin), where multiple comparisons were made using Bonferroni correction, and then the FDR of Benjamini-Hochberg test was applied, with *q*-values < 0.25 considered significant for association of the relative abundances of bacteria with the metadata [27].

### 2.12. Statistical Analyses

Anthropometric, biochemical, diet, and physical activity profiles were analyzed using SPSS v24. For nonparametric data we used the Kruskal-Wallis test followed by Bonferroni correction; for parametric data, we used the ANOVA test, followed by Tukey test (post-hoc assay) to identify significant differences between each set of groups in both cases. To identity differences in gut microbiota at all taxonomic levels among all groups, we analyzed the relative abundance (%) of bacterial communities with the Kruskall-Wallis test to calculate *p*-values; after this, we applied Benjamini-Hochberg correction to determine the *q*-values using p.adjust() function in R software v3.4.2.

## 3. Results

### 3.1. Analysis of Physical Activity, Intake of Nutrients, and Anthropometrical Profile Data Contrast the T2D Groups with the Control and Prediabetes Groups

For this study, we recruited 217 participants of an average age of 49 years, of which 143 were females (Table 1). The anthropometric profile showed that 50% of individuals in all T2D category groups were obese, while 30% of the CO and PRE groups were obese and 48% were overweight. The biochemical profile showed that in the groups diagnosed and treated for T2D, the percentage of patients with HbA1c values above the glycemic goal (>6.5%) for adults were 71% for T2D−M, 68% for T2D−P, and 92% for the T2D−P+I groups. The blood cholesterol level was over the normal value (<200 mg/dL) in PRE, T2D−No−M, and T2D−M groups. The triglycerides levels were elevated in all groups except the T2D−P, which had Bezafibrate as part of the polypharmacy treatment. In general, low density lipoprotein cholesterol was elevated in all groups, with the highest level observed in T2D−No−M, followed by PRE.

The dietary profile applied to all groups showed that daily kilocalorie and macronutrients intake was increased in T2D−P and T2D−P+I individuals followed by the PRE individuals. The daily sugar intake for all groups was in the normal range (<10% of the total calories); however, T2D−P, PRE, and CO groups reported higher intake than the others (Table 1). No significant differences were found among groups regarding physical activity (*p* = 0.526); values indicated sedentary and light physical activity lifestyles for all groups. Comparative analyses of pairs of groups confirmed the differences observed in Table 1 for all the profiles (Appendix A).

### 3.2. Firmicutes and Bacteroidetes Abundance Outline Major Changes in Gut Microbiota Diversity of CO, PRE and T2D Conditions

Bacterial diversity was characterized by high-throughput DNA sequencing of V3-16S rDNA libraries, as described in Materials and Methods. Our results for the PRE and T2D−No−M groups were consistent with other reports showing an increase of Bacteroidetes, and a decrease of Firmicutes at the onset of the disease (Figure 1), although these differences were not statistically significant (Appendix A). On the other hand, we also observed that T2D−P and T2D−P+I groups had similar Firmicutes and Bacteroidetes abundances with respect to the CO group. Significant differences were observed at the phylum level (Firmicutes and Bacteroidetes) in the T2D−M group with respect to the CO and T2D−P groups (Appendix A). In addition, the increase in Firmicute abundance was statistically significant when comparing the T2D−M and T2D−P+I groups (Appendix A), (Figure 1).

### 3.3. The Gut Bacterial Richness and Diversity in the T2D−No−M Group is Lower than the Rest of the Groups

We profiled the gut bacterial communities of the studied individuals measuring Chao1 index as an indicator of richness, and Shannon and Simpson indexes as indicators of diversity (Figure 2). We further performed a comparison analysis of the alpha diversity indexes among groups to detect significant differences (Appendix A). The indexes showed that in terms of richness and diversity, the gut bacterial communities of T2D−No−M individuals were lower than those of CO, PRE, and T2D individuals with medication (T2D−M, T2D−P, and T2D−P+I). There was no detectable difference in the bacterial diversity when data was analyzed by the Simpson index (Appendix A).

### 3.4. There are Significant Differences in Gut Microbiota Beta Diversity between Groups

Beta diversity is a widely accepted parameter to contrast differences in bacterial communities among individuals with different health conditions. Based on data from the OTU table, we made three two-dimensional scatter plots, generated using principal coordinates analysis (PCoA), calculated using unweighted UniFrac analysis and the distances estimated among samples from all groups. The results of this analysis showed no intersubject similarities for individuals belonging to the same category that cluster them apart from the other groups (Figure 3). However, an analysis of similarities (ANOSIM) detected statistical differences in beta diversity between some pairs of groups (Appendix A). Significant differences in beta diversity were observed between CO with T2D−No−M and T2D−M; PRE with T2D−No−M and T2D−M; T2D−M with T2D−P and T2D−P+I; and T2D−No−M with T2D−P, and T2D−P+I (Appendix A). Given that there were more women than men in the study, our observations may be subject to a gender-effect. To evaluate such an effect, we probed our database, making an unweighted UniFrac beta diversity analysis, and found that women and men did not cluster separately (*p* = 0.780) (Appendix A). This could suggest that the gender imbalance in our groups did not have a strong effect on the microbiota. In addition, we made a similar analysis stratifying the participants by age into 37–39, 40–49, 50–59, and 60–63 year-old groups. We found that participants were not clustered separately (*p* = 0.839) (Appendix A).

### 3.5. Differences in Gut Bacterial Abundance are Observed in All Studied Groups

LEfSe analysis was used to identify significant enrichment of gut bacteria among all groups as described in Materials and Methods. We analyzed the six phenotypic categories, grouping the T2D−P and T2D−P+I in one class for analysis. Based on these analyses, for the CO group, we found bacteria from the phylum Proteobacteria, family *Alcaligenaceae*, which were enriched at least 2.9-fold; for the PRE group, the family *Comamonadaceae*, phylum Proteobacteria was enriched at least 2.07-fold; and finally, for the T2D−No−M group, bacteria of the genus *Sutterella*, phylum Proteobacteria were enriched at least 2.85-fold (Figure 4, Appendix A). Patients with T2D and pharmacological treatment showed an increase in bacteria of three different phyla. For T2D−M, the genus *Pelomonas* spp. phylum Proteobacteria was increased 2.61-fold; the order Bacteroidales, phylum Bacteroidetes was increased 4.43-fold; and for the phylum Acidobacteria, the family *Koribacteraceae* was increased 2.78-fold, while the order Acidobacteriales was increased 2.74-fold. Finally, for the combined group T2D−P and T2D−P+I, we found that *Oscillospira* spp. was increased 3.03-fold, and *Roseburia* spp. was increased 2.94-fold, both of which are in the phylum Firmicutes (Figure 4, Appendix A).

### 3.6. Selected Microbial Taxa Are Significantly Associated with Clinical, Anthropometrical, Dietary Intake and Physical Activity Parameters

We searched for variables that were significantly associated with gut microbiota members. From this, we found a positive association of age with the *Enterococcaceae* family (*p* = 0.0001, *q* = 0.0870), and of gender with the genus *Prevotella* (*p* = 0.0002, *q* = 0.0870). Negative associations were found in lipid intake with the genus *Kaistobacter* (*p* = 0.0001, *q* = 0.0870), and in systolic blood pressure with the genus *Erwinia* (*p* = 0.0005, *q* = 0.1692) in all studied subjects (*n* = 217) (Figure 5). We also evaluated statistical associations with variables like body composition and physical activity (*n* = 99) (Figure 5). Positive associations of *Dorea* with physical activity (*p* = 0.0001, *q* = 0.2093), *Enterococcaceae* family with body fat (%) (*p* = 0.0004, *q* = 0.2093), and body fat (kg) (*p* = 0.0003, *q* = 0.2093), and negative association of *Fusobacterium* genus with weight (kg) (*p* = 0.0004, *q* = 0.2093) were observed (Appendix A).

### 3.7. The Predicted Functional Metagenome of the Gut Microbiota Shows Differences in PRE and T2D−M Groups

The prediction of functional genes in the gut microbiota made by PICRUSt showed significant differences in the PRE and T2D−M groups. The results showed an increase in the percent relative frequency of amino sugar and nucleotide sugar metabolism pathways in the resident gut microbiota of the T2D−M group (Figure 6A). To a lesser degree, there was also an increase in energy and amino acid metabolism (glycine, serine and threonine) (Figure 6A), as well as lipid metabolism (sphingolipid), and glycan biosynthesis (lipopolysaccharide biosynthesis) (Figure 6B) in the same group. The resident gut microbiota of the PRE group exhibited an increase in the relative frequency of propanoate metabolism and benzoate degradation (Figure 6B). In addition, there was an increase in genes involved in the biodegradation and metabolism of xenobiotics such as atrazine, dioxin, xylene, and styrene (Figure 6C) (Appendix A).

## 4. Discussion

Type 2 diabetes is a complex multifactorial metabolic disorder, where gut microbiota might be related to the development of this disease. In this work, we performed a cross-sectional study to detect differences in gut microbiota abundance and to assess its potential contribution to T2D among patients undergoing different pharmacological treatments, as well as having been treated for varying time periods. In addition, we studied two interesting groups: prediabetes (PRE) and type 2 diabetes no medication (T2D−No−M) patients that were diagnosed during this study.

We detected, by high-throughput semiconductor DNA sequencing of V3-16S-rDNA libraries prepared from feces, that the relative abundance of the phylum Firmicutes in the T2D−M group, with at least one year of Metformin medication, decreased by almost 50% in comparison to the CO group. This finding contrasts with previously published work, where after two and four months of Metformin treatment, the participants showed an increase in the relative abundance of 86 different bacteria species, mostly Firmicutes [14]. We also found that the alpha and beta diversities of the following groups showed significant statistical differences. For instance, for alpha diversity (number of observed OTUs), diversity is lower in T2D−M compared to CO, T2D−P+I, or PRE; the diversity based on Chao1 index is lower in T2D−M compared to T2D−P, CO, T2D−P+I, and PRE; the diversity in T2D−P is larger compared to T2D−No−M; the diversity in T2D−No−M is lower compared to CO, T2D−P+I, and PRE; and the diversity in T2D−M based on the Shannon index, is lower compared to CO, T2D−P+I and PRE (Appendix A). For the beta diversity, T2D−No−M is lower compared to CO, T2D−P and T2D−P+I; the beta diversity in T2D−M is lower compared to CO, T2D−P+I, and PRE (Appendix A). These results are remarkable, since these individuals were not under any pharmacological treatment, therefore showing how gut microbiota diversity changes at different stages of the disease.

For further analysis, we grouped the individuals of the T2D−P and T2D−P+I groups into a single set, because they had the same pharmacological treatment and treatment duration, with the only difference between them being insulin administration in the latter group. In this combined set, we found two bacteria with statistically significant increases in their abundance, the genus *Oscillospira* (Firmicutes) and the genus *Roseburia*; in contrast, *Roseburia* was decreased in European and Chinese subjects suffering of T2D [11].

The group T2D−M had significant changes in the abundance of the order Bacteroidales (phylum Bacteroidetes), which includes members found increased in the gut microbiota of rats with genetic T2D and nonalcoholic fatty liver disease, but not in patients [28], and decreased in the gut of a normal weight mice model for T2D [29]. On the other hand, members of the group T2D−M have increased fasting glucose levels above 100 mg/dL. This could be explained by the fact that members of the class Acidobacteriia, order Acidobacteriales are reported to have significant saccharolytic activity degrading complex carbohydrates (e.g., cellulose, starch) from the diet, to produce glucose [30], which is absorbed in the colon [31] in part due to Metformin activity [32].

On the other hand, we observed an increase in the abundance of *Pelomonas* spp. in T2D−M but not in T2D−No−M group. A possible interpretation of this finding is that the increase of this bacterium might be due to Metformin treatment rather than T2D per se. Few studies to date have reported information about *Pelomonas* and T2D; however, there is one important report about *Pelomonas puraquae* in which the authors cited evidence suggesting that mothers with T2D expose the fetus to a microbiota abundant in these bacteria [33]. In our study, the group with the smallest number of confounding variables, in terms of pharmacological treatment, was the T2D−No−M group; our results showed that only *Sutterela* spp. was 2-fold more abundant. A similar finding was reported in Danish prediabetic patients, who had a significant increase of this bacterium in comparison to healthy subjects [34] (Figure 4).

The *Commamonadaceae* family, with members typically associated with high-fiber diets and with a contested role in T2D, is reported to be among 22 families contributing to significant differences in the microbial richness between individuals with normal glucose tolerance and those with T2D in subjects of Kazakhs ethnicity [35]. In our study, the *Comamonadaceae* family was found in higher abundance only in members of the PRE group. Moreover, the family *Alcaligenaceae* (order Burkholderiales) in the CO group exhibited a relative abundance, consistent with a healthy metabolic status, as has been reported in studies of healthy human subjects [36] (Figure 4).

*Prevotella* is a bacterium commonly associated with chronic gut inflammation, and has been reported to be less abundant in T2D, in a Chinese population [37]. While the LEfSe analysis did not detect a statistically significant difference in *Prevotella* abundance between groups, this bacterium was among the twenty most abundant gut bacteria in our study. The abundance of these bacteria tends to increase from CO to PRE to T2D−No−M to T2D−M stages, and to decline with polypharmacy and polypharmacy + insulin (T2D−P, T2D−P+I) treatment (Appendix A) (Figure 4).

In this study, we detected an association of gut bacterial abundance with clinical markers, as well as a negative association between systolic blood pressure and *Erwinia* spp. To date, *Erwinia* spp. has not been reported to be associated with diabetes. *Kaistobacter* spp. was found to be negatively associated with lipid intake; however, in previous reports, the same bacteria showed increased abundance in rats with higher levels of Apolipoprotein E. This bacterium has an important role in the metabolism of lipoproteins, especially in the regulation of cholesterol metabolism [38].

In our study, we found some interesting conflicting results in relation to published literature. We observed a positive tendency for physical activity to be associated with *Dorea* abundance (Figure 5H). In a previous study examining elite race walkers with vigorous physical activity, the authors reported an association between *Dorea* abundance and physical activity, where *Dorea* was negatively associated with oxygen consumption in a physical economy test [39]. In the case of the family *Enterococcaceae*, we found positive association with body fat (Figure 5E,F). Contrary to our results, the family *Enterococcaceae* was 24-fold more abundant in the gut microbiome of exercised versus sedentary mice, the latter of which exhibited 42.9% greater body weight and less abundance of this bacterium [40]. In addition, there was a positive correlation between *Enterococcaceae* and age among participants (Figure 5A).

We also explored whether the predicted metabolic pathways in gut bacteria were relevant in this study. We performed a PICRUSt analysis, and found that only the microbiota of PRE and T2D−M conditions had statistically significant differences among relevant metabolic pathways. The more notable differences were in the energy, carbohydrate, amino acid, and lipid metabolism pathways, where their abundance was higher in T2D−M than in the PRE group. This is consistent with the hypothesis that changes in substrate availability in the lumen lead to bacterial abundance changes, as observed in our study. Our observation is supported by previous studies reporting that Metformin improved intestinal glucose sensing, as well as enhanced production of SCFA and gut peptides [41], even when the nutrient intake among members of PRE and T2D−M groups was similar. Currently, there is evidence suggesting that gut microbiota has an active role in the antidiabetic effects of pharmacological treatments [14]; so far, the most studied drug is metformin. Additional clinical studies are needed to investigate the effect of other antidiabetic drugs on gut microbiota.

Lipopolysaccharides or endotoxins are contributing factors that trigger local inflammation in metabolic disorders involving gut microbiota. These compounds are glycolipids found in the outer membrane of gram-negative bacteria, and are implicated in low-grade metabolic inflammation [42]. In a cohort from the FINRISK 97 study [43], patients with T2D had higher endotoxin activity compared to controls. Consistent with our results, the lipopolysaccharide biosynthesis pathway was increased in T2D−M in comparison to the PRE group. This contrasts with a previous study showing that Metformin changes gut microbiota diversity, and reduces intestinal lipid absorption and lipopolysaccharide levels [44].

Another important aspect of our study is the metabolic pathways involved in xenobiotic degradation, which is present in the gut microbiota. Given that oral medications reach the gastrointestinal tract and are exposed to gut microbiota metabolism, in some cases, the efficacy of medications might be altered before being transported to the bloodstream [45,46]. In our study, we did not find significant differences in any metabolic pathway between the T2D−No−M and T2D−M groups. However, we found decreased relative frequency of metabolic pathways related to the degradation of five xenobiotics in T2D−M with respect to the PRE group. These xenobiotic degradation pathways are clinically important due to their effect on drug bioavailability and bioremediation, since many of these pathways play a role in the detoxification of contaminants, such as the Atrazine pathway. Experimental studies, which characterize the removal of xenobiotics by different gut bacteria, are scarce. To date, there have been no previous reports investigating the metabolism of xenobiotics, including Metformin, by the intestinal microbiota from T2D patients. Additional metagenomics studies are required to identify if, in fact, Metformin influences the degradation capacity of the gut microbiota. Limitations of our study include the small size of the T2D−No−M and T2D−M groups, the unequal gender distribution among groups, and the use of only the V3 16S rRNA gene polymorphic region, potentially influencing the determination of the gut microbiota diversity, and hence, the conclusions reported here. Future studies should include a cohort of subjects to monitor disease progression from healthy to prediabetes and to T2D, along with changes in the gut microbiota. In addition, PICRUSt, an important tool used in this study, can only predict known gene families that are included in the orthology reference used.

## 5. Conclusions

In this work, we demonstrated significant differences in gut microbiota diversity among the studied groups, from controls to T2D patients with or without pharmacologic treatment. We observed differences in relative abundance, alpha, and beta diversities, as well as significant associations between specific bacteria and clinical characteristics. Further, we predicted metabolic pathways affected by gut microbiota in relation to T2D−M and PRE patients. Based on the analyses presented in this study, the results indicate that each patient group in our study has specific predicted metabolic characteristics and gut microbiota profiles. The results of this study could be used to help define new therapeutic strategies, to modulate gut microbiota through noninvasive treatments such as dietary intervention, probiotics or prebiotics, and to improve glucose tolerance of both prediabetic individuals and those affected by T2D.

## Figures and Tables

**Figure 1 microorganisms-08-00094-f001:**
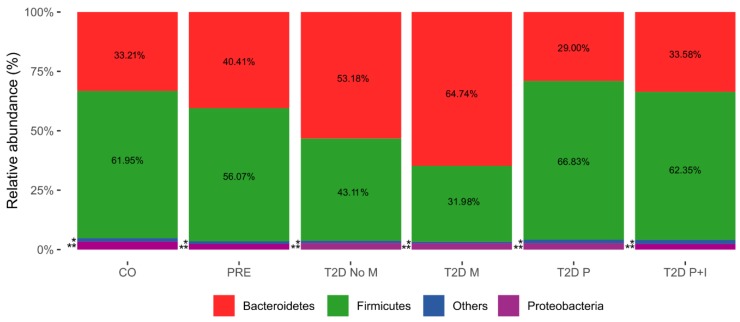
Bacterial phyla abundances. The bar charts of the figure show the relative abundances (%) of relevant bacterial phyla from the gut microbiota in the six groups studied in this work. Control (CO), Prediabetes (PRE), T2D no medicated (T2D−No−M), T2D with Metformin (T2D−M), T2D with polypharmacy (T2D−P), and T2D with polypharmacy + insulin (T2D−P+I). Phyla are identified by colors as indicated underneath the bars. “Others” includes phyla such as Actinobacteria, Tenericutes, Cyanobacteria, and Fusobacteria. One asterisk (*) for Others indicates abundance of 1.50% for CO, 1.10% for PRE, 0.92% for T2D−No−M, 0.63% for T2D−M, 1.51% for T2D−P, and 1.69% for T2D−P+I. Two asterisks (**) for Proteobacteria indicates abundance of 3.34% for CO, 2.42% for PRE, 2.79% for T2D−No−M, 2.65% for T2D−M, 2.66% for T2D−P, and 2.38% for T2D−P+I (see Materials and Methods, and Appendix A).

**Figure 2 microorganisms-08-00094-f002:**
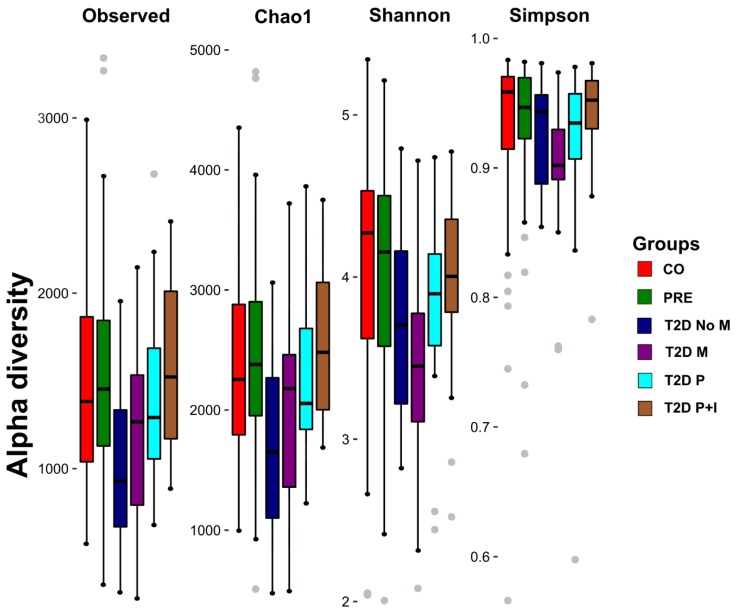
Bacterial alpha diversity. The box-plot figures show the Alpha Diversity of the bacterial communities in the six study groups: Control (CO), Prediabetes (PRE), T2D no medicated (T2D−No−M), T2D with Metformin (T2D−M), T2D with polypharmacy (T2D−P), and T2D with polypharmacy + insulin (T2D−P+I) by means of Observed (number of observed OTUs), Chao1 (richness), and Shannon and Simpson diversity indexes. Plotted in the graphics are the interquartile ranges (bottom and top of boxes), medians (middle lines in the boxes), and the lowest and highest values for the first and third quartiles. Dots represents outlier individuals, and each group is identified by colors as indicated on the legend at the right side of the figure (see Appendix A).

**Figure 3 microorganisms-08-00094-f003:**
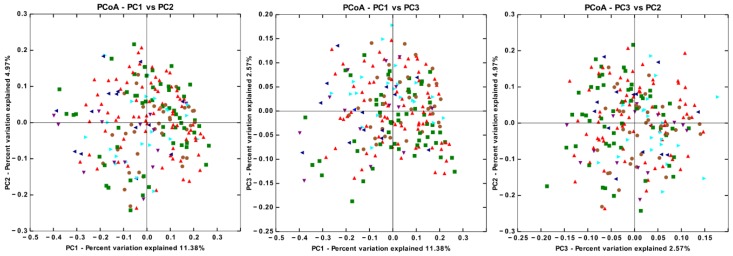
Bacterial beta diversity. Principal Coordinates Analysis based on unweighted UniFrac distances between the gut microbiota profiles of individuals from the six groups. Control (CO) red triangles, Prediabetes (PRE) green squares, T2D no medicated (T2D−No−M) dark blue triangles, T2D with Metformin (T2D−M) purple triangles, T2D with polypharmacy (T2D−P) light blue triangles, and T2D with polypharmacy + insulin (T2D−P+I) brown circles (see Appendix A).

**Figure 4 microorganisms-08-00094-f004:**
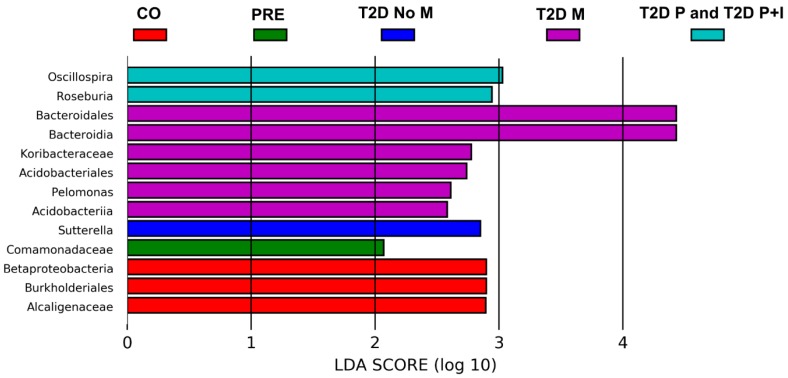
Graphic of the Linear discriminant analysis Effect Size (LEfSe) for the bacterial communities. The LEfSe plot shows enriched bacterial families significantly associated in five groups as follow; 1. Control (CO), 2. Prediabetes (PRE), 3. T2D no medicated (T2D−No−M), 4. T2D with Metformin (T2D−M), and 5. T2D with polypharmacy (T2D−P) and T2D with polypharmacy + insulin (T2D−P+I). Three bacteria were enriched in CO group (red color), one bacteria was enriched in the PRE group (green color), one bacteria was enriched in T2D−No−M group (dark blue color), six bacteria were enriched in the T2D−M group (purple color) and two bacteria in the combined T2D−P and T2D−P+I group (light blue color). The LDA score or Effect size is shown at logarithmic scale underneath the bars. Names of the bacterial families are shown beside the horizontal bars. Each study group is identified by color on top of the figure (see Appendix A).

**Figure 5 microorganisms-08-00094-f005:**
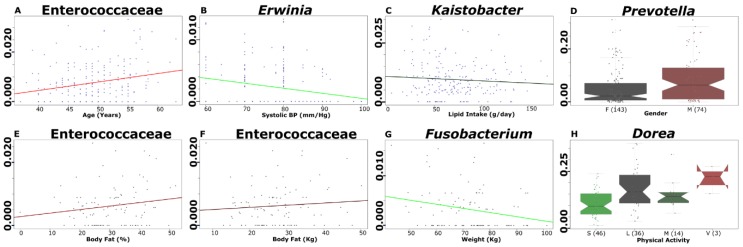
Multivariate association of the gut microbiota with all measured variables using MaAsLin. The figure shows graphs of the significant association between metadata and microbial measurements for 217 individuals of all groups, together with variables such as: age and *Enterococacceae* family (**A**), systolic blood pressure and genus *Erwinia* (**B**), lipid intake and genus *Kaistobacter* (**C**), and gender and genus *Prevotella* (**D**). Additional analyses show significant association between metadata and microbial measurements using a subsample of 99 selected individuals with body composition and physical activity metadata: body fat (%) and family *Enterococacceae* (**E**), body fat (kg) and family *Enterococacceae* (**F**), weight (kg) and genus *Fusobacterium* (**G**), and physical activity and genus *Dorea* (H). In the figure, color lines with positive or negative slopes show the tendency of association; box-plots in the graphic show the interquartile ranges (bottom and top of boxes), medians (middle lines in the boxes), and the lowest and highest values for the first and third quartiles. Dots represents single individuals, and each category is identified by colors. *Y*-axes, indicates relative abundances for each taxon, *X*-axes, indicates the metadata. For Gender, F is 143 females, and M is 74 males; for Physical Activity, S is sedentary (46), L is light (36), M is moderate (14) and V is vigorous (3), numbers in parenthesis indicates de number of participants by group (**D**,**H**) (see Appendix A).

**Figure 6 microorganisms-08-00094-f006:**
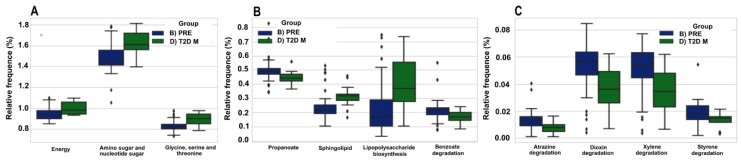
Comparative prediction of the functional metagenome of the gut bacterial microbiota. The figure shows graphic box-plots representations of the proportion of selected predicted sequences using PICRUSt for the bacterial communities. The *Y*-axis shows the relative abundances (%) of the gene content prediction, and the *X*-axis shows PRE (blue color) and T2D−M (green color) groups. The significant metabolic pathways shown at the bottom of each graph are: (**A**) Energy metabolism, amino sugar and nucleotide sugar metabolism, and glycine serine and threonine metabolism; (**B**) Propanoate metabolism, sphingolipid metabolism, lipopolysaccharide biosynthesis, and Benzoate degradation, and (**C**) Atrazine, dioxin, xylene, and styrene degradations. The box-plots show the interquartile ranges (bottom and top of boxes), medians (middle lines in the boxes), and the lowest and highest values for the first and third quartiles. Dots represents single individuals, and each category is identified by colors (see Appendix A).

**Table 1 microorganisms-08-00094-t001:** General characteristics of the study subjects.

	CO	PRE	T2D−No−M	T2D−M	T2D−P	T2D−P+I	*p*-Value
Number of subjects	76	54	14	14	22	37	NA
Gender (women/men)	50/26	36/18	7/7	12/2	14/8	24/13	NA
Age (years)	48.0 ± 5.4	50.2 ± 5.0	48.1 ± 4.7	48.1 ± 4.6	51.3 ± 5.6	50.5 ± 4.5	0.237
Blood Pressure							
SBP (mm/Hg)	75.6 ± 8.3	77.9 ± 9.9	80.4 ± 8.6	77.9 ± 9.6	76.7 ± 7.2	79.0 ± 10.8	0.256
DBP (mm/Hg)	114.5 ± 13.4	117.5 ± 11.3	122.1 ± 10.2	120.5 ± 12.3	117.7 ± 10.3	124.2 ± 28.9	0.078
Anthropometric Profile							
Weight (kg)	71.8 ± 12.9	71.6 ± 11.9	78.9 ± 12.1	73.9 ± 13.6	76.9 ± 15.4	79.4 ± 18.6	0.078
BMI (kg/m^2^)	27.7 ± 3.7	28.4 ± 4.1	30.8 ± 4.6	30.1 ± 4.8	30.3 ± 5.0	31.5 ± 7.8	0.012
Waist Circumference (cm)	89.7 ± 14.0	92.0 ± 11.1	99.3 ± 11.6	97.3 ± 8.3	94.8 ± 20.0	99.8 ± 15.7	0.002
Body Fat Bod Pod (%)	29.6 ± 8.2 ^β^	30.6 ± 9.5 ^δ^	30.8 ± 10.8 ^τ^	34.0 ± 7.8 ^τ^	ND	ND	0.464
Biochemical profile							
HbA1c (%)	5.4 ± 0.2	5.9 ± 0.2	8.4 ± 2.2	7.5 ± 1.0	8.1 ± 2.7	9.4 ± 2.0	<0.001
Fasting glucose (mg/dL)	92.9 ± 10.1	97.2 ± 11.0	199.3 ± 108.4	148.6 ± 40.9	147.7 ± 68.7	186.6 ± 77.5	<0.001
Cholesterol (mg/dL)	195.6 ± 29.3	204.9 ± 53.1	271.0 ± 160.8	205.2 ± 34.3	169.3 ± 31.1 ^Ω^	182.8 ± 35.2 ^β^	<0.001
Triglycerides (mg/dL)	159.6 ± 89.9	154.2 ± 63.2	274.2 ± 191.0	226.6 ± 125.3	148.7 ± 51.1 ^Ω^	206.3 ± 104.6 ^β^	0.002
HDL (mg/dL)	47.5 ± 10.2 ^€^	47.6 ± 9.4 ^¥^	46.9 ± 8.6	45.7 ± 9.5	40.0 ± 7.5 ^Ω^	42.0 ± 8.6 ^π^	0.005
LDL (mg/dL)	114.8 ± 26.2 ^£^	127.4 ± 51.4 ^¥^	137.9 ± 37.5 ^∞^	115.5 ± 29.5 ^α^	102.2 ± 28.2 ^Ω^	105.3 ± 27.9 ^π^	0.019
Dietary profile							
Kcal (kcal/day)	1903.3 ± 678.0	2027.1 ± 608.7	1983.0 ± 610.8	1975.6 ± 780.6	2487.3 ± 811.4	2597.3 ± 887.7	<0.001
Protein (g/day)	90.0 ± 36.1	93.7 ± 31.6	98.1 ± 38.4	98.1 ± 36.0	123.8 ± 45.0	114.6 ± 43.2	0.001
Lipids (g/day)	65.3 ± 29.0	66.6 ± 30.1	63.1 ± 24.9	65.5 ± 23.7	88.5 ± 34.2	92.1 ± 38.5	<0.001
Carbohydrates (g/day)	241.8 ± 94.8	262.8 ± 79.2	256.0 ± 105.4	248.4 ± 122.1	294.2 ± 115.4	317.0 ± 110.4	0.021
Sugar from beverage (g/day)	14.3 ± 22.4	12.4 ± 16.3	7.6 ± 11.4	18.5 ± 19.4	15.7 ± 15.4	15.4 ± 13.8	0.241
Sugar (g/day)	43.8 ± 37.6	34.2 ± 23.9	18.2 ± 18.7	18.5 ± 27.6	30.0 ± 28.9	24.8 ± 23.3	0.011
Physical Activity							
METs	2.4 ± 2.1	2.1 ± 1.9	1.8 ± 0.7	1.4 ± 0.6	1.8 ± 1.3	2.1 ± 2.7	0.526

Results are expressed as mean ± standard deviation. p-value was calculated according to Kruskal Wallis test for nonparametric data and ANOVA for parametric. Control, CO; Prediabetes, PRE; T2D no medication, T2D−No−M; T2D Metformin, T2D−M; T2D polypharmacy, T2D−P; T2D polypharmacy + insulin, T2D−P+I. ND = No determined, NA = Not applicable. High-density lipoprotein cholesterol—HDL, Low-density lipoprotein cholesterol—LDL. β = 35 subjects, *δ* = 36 subjects, *τ* = 14 subjects, € = 61 subjects, £ = 59 subjects, ¥ = 45 subjects, ∞ = 11 subjects, α = 13 subjects, Ω = 21 subjects, π = 33 subjects. Polypharmacy—hypoglycemic agents and other drugs such as antihypertensive, lipid-lowering among others.

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
