# Peer review of "Characterization of the Gut Microbiota of Individuals at Different T2D Stages Reveals a Complex Relationship with the Host"

_microorganisms, 2020, doi:10.3390/microorganisms8010094_

Round 1

Reviewer 1 Report

The authors carried out an observational study on gut microbiota changes in 217 patients at risk or with type 2 diabetes.

Although interesting, I am not convinced that the results presented in this manuscript significantly improve our knowledge in the field. The gender bias is not irrelevant as in different group the ratio female/male is highly diverse. The described significant differences in the studied groups seem to be ascribable to subdominant taxa, and the authors should consider this aspect in the discussion.

Moreover, the manuscript would benefit from better presentation and English language revision.

Major concerns:

- Introduction section: the authors have mentioned previous s studies that report “bacterial dysbiosis or changes” related to T2D. I suggest implementing this paragraph by summarizing the relevant changes already reported in the literature (what taxa have been found altered?)

- Participant recruitment: what are inclusion and exclusion criteria for enrollment? Were the controls (CO) healthy? What samples were collected? It has been not specified.

- Sample collection: were fecal samples collected at the hospital or subjects brought the samples while having a follow-up visit?

- Line 186, page 4: did the authors carried out multiple runs for microbiota sequencing? If this is the case, did they check the results for possible bias related to various run?

- Why fibers have been disregarded in the dietary profile? Moreover, did the authors check for possible correlations between taxa and nutritional parameters? Indeed, Table 1 describes several nutritional differences amongst the studied groups?

- Concerning the alpha-diversity analyses: all statistical differences, but two, encompass T2D-M group. Could the highest number of women in this group (about 86%) be responsible for the result? Hormone-driven differences?

- What are the relative abundances of family/genera found to be altered in different groups (i.e. Oscillospira, Pelonomas, Acidobacteria..)? Are they above 1% in the studied population? Or were these taxa present in all patients within that group?

- Why only PRE and T2D-M predicted functional genome analysis has been reported?

- Discussion section: the authors mentioned statistical differences (line 357) referring to supplementary materials. I suggest summarizing the relevant data.

Minor:

- All taxa but phylum need italics

- The sentence at line 273, page 7, is quite confounding. I suggest rephrasing it.

- In figure 1, the “others” should be at the very bottom (or at the top).

- In figure 3, adding centroids would make the figure interpretation more intuitive.

- Figure 4: I suggest adding p-values to the LEfSe and to shorten the corresponding paragraph in the text (lines 306-320)

Author Response

Manuscript ID: microorganisms-661237

 Title: Characterization of the gut microbiota of individuals at different T2D stages reveals a complex relationship with the host.

 Authors: Alejandra Chávez-Carbajal, María Luisa Pizano-Zárate, Fernando Hernández-Quiroz, Guillermo Federico Ortiz-Luna, Rosa María Morales-Hernández, Amapola De Sales-Millán, María Hernández-Trejo, Angelina García-Vite, Luis Beltrán-Lagunes, Carlos Hoyo-Vadillo, Jaime García-Mena.

 Reviewer 1

Answer: We thank very much the deeply enquiring review of our manuscript made by the reviewer 1. Below we have made our best to reply to all queries to improve our work.

Comments and Suggestions for Authors

The authors carried out an observational study on gut microbiota changes in 217 patients at risk or with type 2 diabetes. Although interesting, I am not convinced that the results presented in this manuscript significantly improve our knowledge in the field.

Answer: we acknowledge the expert opinion of the reviewer; however, we still believe they are novel because we contrasted the diversity of the fecal microbiota and predicted bacterial metabolic pathways, in a sample of individuals at different stages of type 2 diabetes. Most of the multiple reports on gut microbiota and T2D that can be accessed (e.g. PubMed), make comparative studies of one stage with the corresponding controls (Sircana et al., 2018). Moreover, our findings contribute to the scientific knowledge on T2D and gut microbiota in a middle income country like Mexico, with an idiosyncratic population in terms of its ancestry, where the reported prevalence in 2017 was among the highest in the world and it is estimated to soar by year 2045 (Cho et al., 2017). It is also the first time that Mexican subjects are reported in a characterization of the gut microbiota in this wide range of glucose metabolism stages, including subjects that were debutants for T2D and consequently were not medicated. Suitable information has been entered in the Discussion section of the manuscript.

 The gender bias is not irrelevant as in different group the ratio female/male is highly diverse. The described significant differences in the studied groups seem to be ascribable to subdominant taxa, and the authors should consider this aspect in the discussion.

Answer: Regarding the gender imbalance issue in our groups, the reviewer is more than correct drawing our attention to the fact that gender has strong effect on the gut microbiota diversity associated to disease as has been recently reported (Santos-Marcos et al., 2019). In general, we have more women than men in our studied groups and this might be a factor that could bias our results. We probed our database making an Unweighted Unifrac beta diversity analysis and found that women and men did not cluster separately (p = 0.780) (Figure S02). This could suggest that gender unbalance in our groups does not have a strong effect in the microbiota. We entered this information in “3.4. There are significant differences in gut microbiota beta diversity between groups.” part of the Results section lines 451-469. In addition, we report this issue as a limitation of the study indicating that genders in each group were not perfectly matched and this might potentially affect the changes in the microbiota reported in this work, lines 647-665.

Moreover, the manuscript would benefit from better presentation and English language revision.

Answer: We followed the advice of the Reviewer 1 and the manuscript was reviewed by a native English speaker.

Major concerns:

- Introduction section: the authors have mentioned previous s studies that report “bacterial dysbiosis or changes” related to T2D. I suggest implementing this paragraph by summarizing the relevant changes already reported in the literature (what taxa have been found altered?).

Answer: We added the following information “One study in China with 345 individuals determined that people with T2D had a moderate bacterial dysbiosis with respect to the control group with a statistically significant increase in Clostridium bolteae, C. hathewayi, and the family Lachnospiraceae [10]. In another study, with 145 European women over 70 years-old with glucose intolerance, it was determined that the observed changes in the gut microbiota including increase in Clostridium clostridioforme, and decrease in the abundance of Roseburia_272 could be directly related to the development of T2D [11].” in lines 75 to 81.

- Participant recruitment: what are inclusion and exclusion criteria for enrollment? Were the controls (CO) healthy? What samples were collected? It has been not specified.

Answer: the controls were healthy subjects which were classified under the ADA criteria as we described in “2.1 Study subjects” of the “2. Materials and Methods” section lines 113-116. The inclusion criteria were people who decided to participate in the study and provided a blood sample to obtain plasma and feces for study. In the case of T2D patients, those who were diagnosed by parameters established by the American Diabetes Association and who were between 40 and 55 years old were included. The exclusion criteria were gastrointestinal diseases, antibiotic treatment during the last 3 months, vegetarians and vegans, allergies and autoimmune disease. This information was added in lines 121-126. The type of samples collected were described in each subsection of the of the “2. Materials and Methods” section for the Biochemical profile and DNA extraction from feces.

- Sample collection: were fecal samples collected at the hospital or subjects brought the samples while having a follow-up visit?

Answer: As it is mentioned in the manuscript in the “2.6. DNA extraction from feces” of the “2. Materials and Methods” section, lines 194-195. Fecal samples were obtained from the participants at the hospital and immediately stored at -70° C until DNA was extracted.

 - Line 186, page 4: did the authors carried out multiple runs for microbiota sequencing? If this is the case, did they check the results for possible bias related to various run?

Answer: The 217 samples were sequenced using five Ion Torrent 318 chips in pools of 50 individuals including one additional sample (internal control), a preparation of DNA of 10 different bacteria, which is used every time we sequence in our laboratory. There was not statistically significant difference when we compared the results.

- Why fibers have been disregarded in the dietary profile? Moreover, did the authors check for possible correlations between taxa and nutritional parameters? Indeed, Table 1 describes several nutritional differences amongst the studied groups?

Answer: In the2.3. Dietary assessment” of the “2. Materials and Methods” section, lines 149-155 we refer the use of the Mexican food composition list “Sistema Mexicano de Equivalentes”; this reference is the closest reference with suitable nutritional information about specific Mexican food and products. This is the reason why we chose this reference and no other food list for other country or population. This list has the limitation that among all included food groups, only fruits and vegetables have the fiber content. For this reason, we only show the macronutrients list in Table 1. Regarding the several nutritional differences among the studied groups, we did MaAsLin analysis of data of all nutrients and all bacteria to identify possible associations; finding that only Kaistobacter spp. was associated with lipid intake with statistical significance (Fig. 5).

- Concerning the alpha-diversity analyses: all statistical differences, but two, encompass T2D-M group. Could the highest number of women in this group (about 86%) be responsible for the result? Hormone-driven differences?

Answer: The reviewer is right, although as we mention above we probed our database making an Unweighted Unifrac beta diversity analysis and found that women and men did not cluster separately (p = 0.780) (Figure S02), suggesting that gender unbalance in our groups does not have a strong effect in the microbiota, we completely agree that for the T2D-M group where women account for about 86%, the gender imbalance in the sample might have a relevant influence in the result for alpha-diversity determination. We believe this is a limitation of the study which is acknowledged indicating that genders in each group were not perfectly matched and this might potentially affect the changes in the microbiota reported in this work, lines 647-660.

- What are the relative abundances of family/genera found to be altered in different groups (i.e. Oscillospira, Pelonomas, Acidobacteria..)? Are they above 1% in the studied population? Or were these taxa present in all patients within that group?

Answer: In “3.5. Differences in gut bacterial abundance at family and genera level are observed in all studied groups” of the “3. Results” section, we show in Figure 4 the results of the LEfSe analysis, which is used to identify significant enrichment of gut bacteria among samples. This analysis does not report relative abundances but LDA scores for bacteria with statistically significant differences among the compared groups. For CO, the relative abundance of the family Alcaligenaceae was 0.1934%, order Burkholderiales 0.0007%, and class Betaproteobacteria 0.00001%; for PRE the relative abundance of the family Comamonadaceae was 0.0747%; for T2D−No−M, the relative abundance of genus Sutterella was 0.9734%; for T2D−M, the relative abundance of the class Acidobacteriia, order Acidobacteriales, family Koribacteraceae was 0.0003%, for genus Pelomonas was 0.0004%, for the class Bacteroidia, order Bacteroidales was 0.4185%; finally for T2D−P and T2D−P+I, the abundance was genus Roseburia 2.89% and genus Oscillospira, 3.24%. This information was entered at the bottom of Table S5.

- Why only PRE and T2D-M predicted functional genome analysis has been reported?

Answer: In 4. Discussion section, we explain that “We explored also if the predicted metabolic pathways from the gut bacteria were relevant in this study. We made PICRUSt analysis and found that only the microbiota of PRE and T2D−M conditions had statistically significant differences among relevant metabolic pathways”.

- Discussion section: the authors mentioned statistical differences (line 357) referring to supplementary materials. I suggest summarizing the relevant data.

Answer: We summarized the relevant data as follows “We also found that the alpha- and beta-diversities of the following groups showed significant statistical differences. For instance for alpha diversity, for Observed, T2D−M with CO, T2D−P+I, and PRE; for Chao1 index, T2D−M with T2D−P, CO, T2D−P+I, and PRE, T2D−P with T2D−No−M, and CO with T2D−No−M; and for Shannon index T2D−M with CO, T2D−P+I and PRE (Table S3). For beta diversity, T2D−No−M with CO, T2D−P and T2D−P+I; T2D−M with CO, T2D−P+I and PRE (Table S4).” This information was entered in lines 529-534.

Minor:

- All taxa but phylum need italics

Answer: all taxa are written in italics at the level of family and below in the text and tables (https://wwwnc.cdc.gov/eid/page/scientific-nomenclature)

- The sentence at line 273, page 7, is quite confounding. I suggest rephrasing it.

Answer: The paragraph was rephrased to “Bacterial diversity was characterized by high-throughput DNA sequencing of V3-16S rDNA libraries as described in Materials and Methods. Our results for the PRE and T2D−No−M groups were consistent with other reports showing an increase of Bacteroidetes, and a decrease of Firmicutes at the onset of the disease (Fig. 1), although these differences were not statistically significant (Table S2).” and the text was entered at lines 281-284 in “3.2. Firmicutes and Bacteroidetes abundance outline major changes in gut microbiota diversity of CO, PRE, and T2D conditions.” of “3. Results” section.

- In figure 1, the “others” should be at the very bottom (or at the top).

Answer: the Figure 1 was modified as requested and placed in the text.

- In figure 3, adding centroids would make the figure interpretation more intuitive.

Answer: we agree with the reviewer; however, we used the program make_2d_plots.py from the QIIME pipeline and it does not offer this option.

- Figure 4: I suggest adding p-values to the LEfSe and to shorten the corresponding paragraph in the text (lines 306-320).

Answer: The values noted for each bacterium in the paragraph “3.5. Differences in gut bacterial abundance at family and genera level are observed in all studied groups.” were removed, and the p- and q-values corresponding to the Figure. 4 (LEfSe analysis) are tabulated in the Table S5.

References for Reviewer 1.

 Cho NH, Shaw JE, Karuranga S, Huang Y, da Rocha Fernandes JD, Ohlrogge AW, et al. IDF Diabetes Atlas: Global estimates of diabetes prevalence for 2017 and projections for 2045. Diabetes Res Clin Pract. 2018; 138:271-281.

Santos-Marcos et al., Sex Differences in the Gut Microbiota as Potential Determinants of Gender Predisposition to Disease. Mol Nutr Food Res. 2019 Apr;63(7):e1800870. doi: 10.1002/mnfr.201800870. Epub 2019 Feb 13.

Sircana A, Framarin L, Leone N, Berrutti M, Castellino F, Parente R, De Michieli F, Paschetta E, Musso G. Altered Gut Microbiota in Type 2 Diabetes: Just a Coincidence? Curr Diab Rep. 2018 Sep 13;18(10):98.

 --end of text--

Reviewer 2 Report

Here the authors characterized the gut microbiota of 217 Mexican subjects divided into the following six groups: control, prediabetes, type 2 diabetes without medication, type 2 diabetes with metformin, type 2 diabetes with polypharmacy and type 2 diabetes with polypharmacy and insulin, by 16S rRNA gene-based NGS and inferred metagenomics (i.e. PICRUSt).

The paper is overall well written but some parts, especially the microbiota one need some adjustments (it is evident that the authors are not microbiota experts).

In particular:

The classification in gram negative and positive is simplistic and not entirely correct (L60-61). NGS and bioinformatics. The entire section lacks important details. Please describe the PCR mixture and thermocycle, the clean-up and indexing steps, pool concentration, chimera filtering, OTU determination, etc. Furthermore, the authors chose the V3 region that is not widely used on the international scene (as compared to V3-V4), making their data barely comparable (please, highlight this limit). Predicted metagenomics. The authors should tone down any statements related to the gut microbiome functionality since, as they state, PICRUSt is a prediction tool and as such, it has some limitations. Eg. L42 and 455: the authors cannot state that “each group has specific metabolic characteristics”. Paragraph 3.1. The title does not reflect the content of the paragraph (light physical activity concerns every group, high intake of nutrients also concerns prediabetes subjects). Please rewrite. Differences in microbiota beta diversity. The authors state that “the results … showed no inter-subject similarities for individuals belonging to the same category” but shortly after, they write that “ANOSIM data analysis detected statistical differences …”. On what was their first statement based, if not on a statistical test? Family and genus-level differences. The authors find differences that are generally not consistent with the available literature. Please, further discuss, also the possible reason for the discrepancies. Just as an example, what about Collinsella that is typically associated with metabolic disorders? What about the impact of metformin that is well established in the literaure? What is the relative abundance of significantly different taxa? Some genera are usually poorly represented in the human gut microbiota. Furthermore, the title of the paragraph 3.5 does not reflect its content since data at phylum, order, etc. are shown and discussed as well.

Additional comments:

English edits are needed throughout the manuscript. Please use italics for bacterial genera. Abstract: please specify the number of subjects for each group. L202: phyloseq Please add refs for R packages L204: please specify the distance metrics. L212-213: please rephrase (OTUs level genera assignment makes little sense). L219: “predictive functional gene” makes no sense. Please rephrase. L222: please add the ref for PICRUSt. L245: phylum instead of phyla L251: 50 or 49 years old, as in L115? L256: 7.0 or 6.5%? Please be consistent throughout the manuscript. L257: cholesterol or glucose? L292: PCoA stands for Principal Coordinates Analysis What about weighted UniFrac data? Please rephrase the legend of Figure 3. Acidobacteria and not Acidobacteriia

Author Response

Manuscript ID: microorganisms-661237

 Title: Characterization of the gut microbiota of individuals at different T2D stages reveals a complex relationship with the host.

 Authors: Alejandra Chávez-Carbajal, María Luisa Pizano-Zárate, Fernando Hernández-Quiroz, Guillermo Federico Ortiz-Luna, Rosa María Morales-Hernández, Amapola De Sales-Millán, María Hernández-Trejo, Angelina García-Vite, Luis Beltrán-Lagunes, Carlos Hoyo-Vadillo, Jaime García-Mena.

 Reviewer 2

Answer: We thank the review for the careful review of our manuscript that give us the opportunity of learning and improving our work. In the following lines we made our best to answer all the queries.

Comments and Suggestions for Authors

Here the authors characterized the gut microbiota of 217 Mexican subjects divided into the following six groups: control, prediabetes, type 2 diabetes without medication, type 2 diabetes with metformin, type 2 diabetes with polypharmacy and type 2 diabetes with polypharmacy and insulin, by 16S rRNA gene-based NGS and inferred metagenomics (i.e. PICRUSt). The paper is overall well written but some parts, especially the microbiota one need some adjustments (it is evident that the authors are not microbiota experts).

Answer: we thank the encouraging remarks of the reviewer.

In particular:

-The classification in gram negative and positive is simplistic and not entirely correct (L60-61).

Answer: we agree with the reviewer, and the reference to the gram (+/-) was removed from the text in lines 60-61.

-NGS and bioinformatics. The entire section lacks important details. Please describe the PCR mixture and thermocycle, the clean-up and indexing steps, pool concentration, chimera filtering, OTU determination, etc.

Answer: we performed NGS using Ion Torrent technology, and consequently used “barcodes” to identify samples and not “indexes” as is used in the Illumina technology. Regarding the details of the PCR we have that an amplicon of approximately 281 bp containing the V3 variable region of the 16S RNA gene was amplified using V3‒341F forward primer (set of barcodes) complementary to positions 340–356 of the Escherichia coli 16S rDNA molecule rrnB GenBank J01859.1, and the V3‒518R reverse primer complementary to positions 517–533. The PCR mixture was 1X Buffer, 2 mM MgCl2, 0.2 mM of dNTP’s, 0.2 μM 16S rRNA forward barcoded and reverse primers, and 20 ng of template DNA in a final volume of 50 μl. The thermocycler programming was: 5 minutes at 95° C; 25 cycles of 15 seconds at 94° C, 15 seconds at 62° C and 15 seconds at 72° C; subsequently 10 minutes at 72° C. The amplification of the DNA of each sample was performed by PCR GeneAmp System 2700 Thermocycler (Applied Biosystems). After this PGM sequencing was made in-house using Ion OneTouch 2, Ion PGM Template OT2 200 Kit v2 DL (Life Technologies, California, USA), Ion 318 Chip Kit v2 and Ion Torrent PGM System. After sequencing, reads were filtered by the PGM software to exclude low quality and polyclonal sequences. All reads were trimmed to 200 nt length using FastQC. Filtered and demultiplexed FASTQ files were converted into FASTA files, concatenated into a single file and then processed with multiple QIIME (Quantitative Insights into Microbial Ecology) v1.9.0 scripts (http://qiime.org/scripts/)”, this information was added to the text in lines 204-222.

 -Furthermore, the authors chose the V3 region that is not widely used on the international scene (as compared to V3-V4), making their data barely comparable (please, highlight this limit).

Answer: we acknowledge ye expert opinion of the reviewer, however we have use successfully the V3 16S rRNA gene region for the characterization of the bacterial diversity in our published studies (e.g. Hernández-Quiroz et al., 2019; Chávez-Carbajal et al., 2019; Nirmalkar et al., 2018; Murugesan et al., 2015). It is important to mention that although recent published systematic work on bacterial diversity characterization based on the 16S rRNA gene variable regions, indicates that V4 is the most prominent V region for achieving good domain specificity, higher coverage and a broader spectrum in the bacteria domain (Zhang et al., 2018); same work also states that single V3, V4 and V6 regions are also reliable and comparable for the overall coverage of the phylum to genus level. We acknowledge this issue in the “limitations” lines 647-662.

-Predicted metagenomics. The authors should tone down any statements related to the gut microbiome functionality since, as they state, PICRUSt is a prediction tool and as such, it has some limitations. Eg. L42 and 455: the authors cannot state that “each group has specific metabolic characteristics”.

Answer: we modified the text in line 43 to “each group has specific predicted metabolic characteristics and gut bacteria populations for each phenotype”; and the text of lines 671-673 to “Based on the analyses presented in this study, these results indicate that each patient group in our study has specific predicted metabolic characteristics and gut microbiota profiles”.

-Paragraph 3.1. The title does not reflect the content of the paragraph (light physical activity concerns every group, high intake of nutrients also concerns prediabetes subjects). Please rewrite.

Answer: we rephrased the title to 3.1. The analysis of physical activity, intake of nutrients, and anthropometrical profile data contrast the T2D groups with the Control and Prediabetes groups.

-Differences in microbiota beta diversity. The authors state that “the results … showed no inter-subject similarities for individuals belonging to the same category” but shortly after, they write that “ANOSIM data analysis detected statistical differences …”. On what was their first statement based, if not on a statistical test?

Answer: we agree with the reviewer and rephrased the paragraph for clarity as follows “The results of this analysis showed no inter-subject similarities for individuals belonging to the same category that cluster them apart from the other groups (Fig. 3). However, an Analysis of similarities (ANOSIM) detected statistical differences in beta-diversity between some pairs of groups (table S4).” lines 456-457.

-Family and genus-level differences. The authors find differences that are generally not consistent with the available literature. Please, further discuss, also the possible reason for the discrepancies. Just as an example, what about Collinsella that is typically associated with metabolic disorders?

Answer: : Our intention in the Discussion was to put our results in context with respect to the knowledge in the field; in addition, we wanted to get the attention of the reader to other important works where the results not necessarily agree with our results. In some cases, there are not comparable publications on a particular bacterium reported significantly increased or decreased in our work, in these cases we searched for the closest example (report) in other diseases or models. Regarding Collinsella spp., we are aware of the reports about this interesting bacterium associated with atherosclerosis found in the gut of habitants of Marseille (France) (Dione et al. 2018), and whose relative abundance decreases significantly in the gut microbiota of German participants of a weight-loss program suffering of obesity plus type 2 diabetes (Frost et al., 2019). We detected Collinsella spp. in the fecal samples of individuals of all the groups studied in our work; however, the relative abundance did not show statistically significant change in the T2D stages with respect to the Controls.

-What about the impact of metformin that is well established in the literaure? What is the relative abundance of significantly different taxa? Some genera are usually poorly represented in the human gut microbiota. Furthermore, the title of the paragraph 3.5 does not reflect its content since data at phylum, order, etc. are shown and discussed as well.

Answer: We commend the remark of the reviewer about Metformin. Patients in the T2D-M group were medicated with oral Metformin for at least 1-year. This information is mentioned in “Study subjects” of the Materials and Methods section, and the Discussion section. We also say that members of the group T2D−M have increased fasting glucose levels due to the increase in the abundance of bacteria belonging to the class Acidobacteriia, order Acidobacteriales because such bacteria are reported to have significant saccharolytic activity degrading diverse complex carbohydrates present in the diet like cellulose, and starch producing glucose; this glucose is absorbed in the colon [Long et al., 1967] along with glucose which reaches the large intestine as consequence of the reduction in glucose absorption in the small intestine due to Metformin activity [Wu et al., 2017]. This information is mentioned in the Discussion section. The title of paragraph 3.5 was rephrased to “3.5. Differences in gut bacterial abundance are observed in all studied groups”.

Additional comments:

-English edits are needed throughout the manuscript.

Answer: the manuscript originally submitted and reviewed in the first round, had been already edited by a native English-speaking scientist; in addition, the English of this version was also reviewed.

-Please use italics for bacterial genera.

Answer: this was made throughout the text.

-Abstract: please specify the number of subjects for each group.

Answer: the number of subjects for each category are specified in Table 1 and they have been added to the Abstract.

-L202: phyloseq Please add refs for R packages

Answer: the requested reference is McMurdie P J, Holmes S. phyloseq: An R Package for Reproducible Interactive Analysis and Graphics of Microbiome Census Data. PLoS One. 2013; 8(4): e61217. Published online 2013 Apr 22. doi: 10.1371/journal.pone.0061217 and was added in “2.8. Microbial diversity analysis” line 224.

 -L204: please specify the distance metrics.

Answer: when we mention “we used the compare_categories.py script of QIIME, using a distance matrix as the primary input and mapping file” we make reference to a file containing multiple numerical data after comparing the OTUs of all individuals.

-L212-213: please rephrase (OTUs level genera assignment makes little sense).

Answer: we modified the text to “The LEfSe analysis was calculated with the relative abundance information file of all bacteria obtained at Class, Order, Family and Genus taxonomical level.” line 242.

-L219: “predictive functional gene” makes no sense. Please rephrase.

Answer: text was rephrased to “2.10. Predictive functional metagenome of the gut microbiota using PICRUSt” in line 249.

-L222: please add the ref for PICRUSt.

Answer: PICRUSt is properly referenced like “(http://galaxy.morganlangille.com)” in line 254.

-L245: phylum instead of phyla

Answer: change was made at line 340.

-L251: 50 or 49 years old, as in L115?

Answer: data was changed to 49 in line 346.

-L256: 7.0 or 6.5%? Please be consistent throughout the manuscript.

Answer: the value was changed to 6.5 % in line 350.

-L257: cholesterol or glucose?

Answer: it is Cholesterol, data was updated to < 200 mg/dl in line 352.

-L292: PCoA stands for Principal Coordinates Analysis

Answer: correction was made in line 454.

-What about weighted UniFrac data? Please rephrase the legend of Figure 3.

Answer: in the legend of Figure 3, we report results of an unweighted UniFrac analysis.

-Acidobacteria and not Acidobacteriia

Answer: Acidobacteriia is correct since we are referring to the Class Acidobacteriia.

References for Reviewer 2.

Chávez-Carbajal A, Nirmalkar K, Pérez-Lizaur A, Hernández-Quiroz F, Ramírez-Del-Alto S, García-Mena J, Hernández-Guerrero C. Gut Microbiota and Predicted Metabolic Pathways in a Sample of Mexican Women Affected by Obesity and Obesity Plus Metabolic Syndrome. Int J Mol Sci. 2019 Jan 21;20(2).

Dione N, Ngom II, Valles C, Cadoret F, Fournier PE, Raoult D, Lagier JC. 'Collinsella provencensis' sp. nov., 'Parabacteroides bouchesdurhonensis' sp. nov. and 'Sutterella seckii,' sp. nov., three new bacterial species identified from human gut microbiota. New Microbes New Infect. 2018 Feb 22;23:44-47. doi: 10.1016/j.nmni.2018.02.003. eCollection 2018 May.

Frost F, Storck LJ, Kacprowski T, Gärtner S, Rühlemann M, Bang C, Franke A, Völker U, Aghdassi AA, Steveling A, Mayerle J, Weiss FU, Homuth G, Lerch MM. A structured weight loss program increases gut microbiota phylogenetic diversity and reduces levels of Collinsella in obese type 2 diabetics: A pilot study. PLoS One. 2019 Jul 18;14(7):e0219489. doi: 10.1371/journal.pone.0219489. eCollection 2019.

Hernández-Quiroz F, Nirmalkar K, Villalobos-Flores LE, Murugesan S, Cruz-Narváez Y, Rico-Arzate E, Hoyo-Vadillo C, Chavez-Carbajal A, Pizano-Zárate ML, García-Mena J. Influence of moderate beer consumption on human gut microbiota and its impact on fasting glucose and β-cell function. Alcohol. 2019 Jun 12. pii: S0741-8329(19)30068-0. doi: 10.1016/j.alcohol.2019.05.006. [Epub ahead of print].

Long C.L.; Geiger J.W.; Kinney J.M. Absorption of glucose from the colon and rectum. Metabolism 1967, 16, 413-8. doi: 10.1016/0026-0495(67)90132-1.

McMurdie P J, Holmes S. phyloseq: An R Package for Reproducible Interactive Analysis and Graphics of Microbiome Census Data. PLoS One. 2013; 8(4): e61217. Published online 2013 Apr 22. doi: 10.1371/journal.pone.0061217.

Murugesan S, Ulloa-Martínez M, Martínez-Rojano H, Galván-Rodríguez FM, Miranda-Brito C, Romano MC, et al. Study of the diversity and short-chain fatty acids production by the bacterial community in overweight and obese Mexican children. Eur J Clin Microbiol Infect Dis. 2015;34: 1337-1346.

Nirmalkar K, Murugesan S, Pizano-Zárate ML, Villalobos-Flores LE, García-González C, Morales-Hernández RM, Nuñez-Hernández JA, Hernández-Quiroz F, Romero-Figueroa MDS, Hernández-Guerrero C, Hoyo-Vadillo C, García-Mena J. Gut Microbiota and Endothelial Dysfunction Markers in Obese Mexican Children and Adolescents. Nutrients. 2018 Dec 19;10(12).

Wu T.; Xie C.; Wu H.; Jones K.L.; Horowitz M.; Rayner C. Metformin reduces the rate of small intestinal glucose absorption in type 2 diabetes. Diabetes Obes Metab 2017, 19, 290-293. doi: 10.1111/dom.12812.

Zhang J, Ding X, Guan R, Zhu C, Xu C, Zhu B, Zhang H, Xiong Z, Xue Y, Tu J, Lu Z. Evaluation of different 16S rRNA gene V regions for exploring bacterial diversity in a eutrophic freshwater lake. Sci Total Environ. 2018 Mar 15;618:1254-1267. doi: 10.1016/j.scitotenv.2017.09.228. Epub 2017 Oct 28.

--end of text--

Reviewer 3 Report

The authors investigated the differences in the gut microbial diversity in human subjects at different stages of type 2 diabetes (T2D) development, with and without pharmacological treatment. The results suggest considerable changes in microbial diversity in patients with T2D and how different medications could impact such changes. The data is encouraging, and the paper is generally well-written. The data will be helpful in exploring the diet-based therapeutics to counteract the negative impacts of medications and disease severity. 

The meticulousness of the methodology clearly led to the clarity of the results and aids the quality discussion. The data analysis tools including microbiota-analysis are judiciously utilized to interpret and visualize the results.

In the microbial analysis though, the authors have given more importance to Firmicutes and Bacteroidetes abundance. Although these two groups are important, they are a component of a complex network of the microbiota. Other phyla and genera are as important in understanding their role in the disease. The supplementary figure S1 should be included in the manuscript and discuss the relevance of other bacterial members that were affected in these T2D patients (with/without the types of medications and pre-diabetic stage) compared to the healthy individuals. Effect on the abundance of members of Ruminococcaceae, Clostridiales, Lachnospira among others needs to be interpreted discussed in the manuscript in relevance to diabetes and medications.

Author Response

Manuscript ID: microorganisms-661237

 Title: Characterization of the gut microbiota of individuals at different T2D stages reveals a complex relationship with the host.

 Authors: Alejandra Chávez-Carbajal, María Luisa Pizano-Zárate, Fernando Hernández-Quiroz, Guillermo Federico Ortiz-Luna, Rosa María Morales-Hernández, Amapola De Sales-Millán, María Hernández-Trejo, Angelina García-Vite, Luis Beltrán-Lagunes, Carlos Hoyo-Vadillo, Jaime García-Mena.

 Reviewer 3

Answer: We thank very much the review made to our manuscript by the reviewer 3. Below we have made our best to reply to all queries to improve our work.

Comments and Suggestions for Authors

The authors investigated the differences in the gut microbial diversity in human subjects at different stages of type 2 diabetes (T2D) development, with and without pharmacological treatment. The results suggest considerable changes in microbial diversity in patients with T2D and how different medications could impact such changes. The data is encouraging, and the paper is generally well-written. The data will be helpful in exploring the diet-based therapeutics to counteract the negative impacts of medications and disease severity. The meticulousness of the methodology clearly led to the clarity of the results and aids the quality discussion. The data analysis tools including microbiota-analysis are judiciously utilized to interpret and visualize the results.

-In the microbial analysis though, the authors have given more importance to Firmicutes and Bacteroidetes abundance. Although these two groups are important, they are a component of a complex network of the microbiota. Other phyla and genera are as important in understanding their role in the disease.

Answer: we completely agree with the reviewer, however, in our study at phyla level, only Firmicutes and Bacteroidetes showed statistically significant changes at the different T2D stages as is reported and showed in Table S2. However, we detected statistically significant changes for bacteria such as the ones shown in the LEfSe analysis (Figure 4, Table S5) and MaAsLin analysis (Figure 5, Table S6).

-The supplementary figure S1 should be included in the manuscript and discuss the relevance of other bacterial members that were affected in these T2D patients (with/without the types of medications and pre-diabetic stage) compared to the healthy individuals. Effect on the abundance of members of Ruminococcaceae, Clostridiales, Lachnospira among others needs to be interpreted discussed in the manuscript in relevance to diabetes and medications.

Answer: On regard of the bacteria where the relative abundance shows a tendency of change in the Figure S1, in the main text we discussed only the ones where there was additional information about statistically significant changes as Oscillospira (LEfSe), Roseburia (LEfSe), Prevotella (MaAsLin) and Dorea (MaAsLin). For this reason we believe the addition of the Figure S1 to the main text does not improve the Results presentation.

--end of text--

Round 2

Reviewer 1 Report

The authors somehow improved the manuscript, but several issues are still present:

-          English language and punctuation;

-          The reported inclusion criteria are unsuitable: “The inclusion criteria were people who decided to participate in the study and provided a blood sample to obtain plasma and feces for study. Moreover, what subjects were enrolled in the control group? Healthy controls? If so, why the mean BMI is in the overweight range? Was a specific BMI range considered an inclusion criterium? The number of patients/group is highly diverse (76 CO, 54 PRE, 14 T2D−No−M, 14 T2D−M, 22 T2D−P, 37 T2D−P I), impacting on the statistical power of the study. What was the experimental plan?

-          Dietary macronutrient analysis needs to be performed using scientific or international Nutrient Profiling Systems

-          Post-sequence analysis: the sample reads range from a minimum of 10,907 and a maximum of 382,307 reads: how did the authors perform the normalization? It seems a very huge difference. Was the patient sample with 10,907 reads excluded?

Author Response

2019-12-29 Sunday

Manuscript ID: microorganisms-661237

Title: Characterization of the gut microbiota of individuals at different T2D stages reveals a complex relationship with the host.

Authors: Alejandra Chávez-Carbajal, María Luisa Pizano-Zárate, Fernando Hernández-Quiroz, Guillermo Federico Ortiz-Luna, Rosa María Morales-Hernández, Amapola De Sales-Millán, María Hernández-Trejo, Angelina García-Vite, Luis Beltrán-Lagunes, Carlos Hoyo-Vadillo, Jaime García-Mena.

Reviewer 1

Comments and Suggestions for Authors

The authors somehow improved the manuscript, but several issues are still present:

English language and punctuation;

Answer: Thanks for the careful revision of our work, all the English language and punctuation was checked.

-The reported inclusion criteria are unsuitable: “The inclusion criteria were people who decided to participate in the study and provided a blood sample to obtain plasma and feces for study. Moreover, what subjects were enrolled in the control group? Healthy controls? If so, why the mean BMI is in the overweight range? Was a specific BMI range considered an inclusion criterium?

Answer: As me mention in 3.1 of the Results section, the anthropometric profile showed that 50% of individuals in all T2D category groups were obese, while 30% of individuals of the CO and PRE groups were obese and 48% were overweight. The 76 Mexican subjects in the healthy (CO) group are 48.0±5.4 years old and they have an average BMI (kg/m2) of 27.7± 3.7. As the reviewer certainly points out, they are in the overweight range (25 and 29.9) which is common for nowadays Mexican population at this age. We did not find many individuals in the healthy weight range (18.5 and 24.9). Moreover, in this study, we explored the role of the gut microbiota on T2D, no obesity; for this reason, the individuals in the CO group have fasting glucose and HbA1c within normal range.

-The number of patients/group is highly diverse (76 CO, 54 PRE, 14 T2D−No−M, 14 T2D−M, 22 T2D−P, 37 T2D−P I), impacting on the statistical power of the study. What was the experimental plan?

Answer: The plan was not to have those disparate numbers in the groups; we recruited more individuals than described; however, we report the number of participants that remained after applying the exclusion criteria at the moment of recruiting them. We even faced that some of them retired from the study, while others were even eliminated from the study due to the use of oral antibiotics. All of the statistical analyses were made according to the number of individuals by group and kind of data (anthropometric or non-anthropometric data), description of the analyses are under each table with statistical data.

-Dietary macronutrient analysis needs to be performed using scientific or international Nutrient Profiling Systems.

Answer: We thank the reviewer for bringing out this issue which allows improving our manuscript. We did not explain appropriately that we framed the nutrients of the food Mexicans consume, with information taken from the FoodData Central, U.S. Department of Agriculture (https://fdc.nal.usda.gov/). Based on this, we assessed the nutrients consumed by participants in each category. It is reported that the publication “Sistema Mexicano de Alimentos Equivalentes”[17., Pérez Lizaur et al, 2014] has more than 70 citations in the specialized scientific literature. (https://www.researchgate.net/publication/309230928_Sistema_Mexicano_de_Alimentos_Equivalentes).

The text in the manuscript: “2.3. Dietary assessment. A registered dietitian performed a “24-hour Dietary Recall” to assess detailed information about all food and beverages consumed in one day. We used the Mexican food composition list “Sistema Mexicano de Equivalentes” [17], to calculate the daily intake of total kilocalories, macronutrients, and sugar.”

Was changed to: “2.3. Dietary assessment. A registered dietitian performed a “24-hour Dietary Recall” to assess detailed information about all food and beverages consumed in one day. We calculated the daily intake of total kilocalories, macronutrients and sugar with the use of data about nutrients, measures, and other related information from the FoodData Central, U.S. Department of Agriculture (https://fdc.nal.usda.gov/) and the Mexican Food Equivalent Composition List “Sistema Mexicano de Equivalentes” [17].”

-Post-sequence analysis: the sample reads range from a minimum of 10,907 and a maximum of 382,307 reads: how did the authors perform the normalization? It seems a very huge difference. Was the patient sample with 10,907 reads excluded?

Answer: There is a 35-fold difference between the uppermost value and the lower value, however, the mean was 120,947 reads per sample. We used samples with more than 10,000 Ion Torrent reads. We did not normalize the number to one set value since the rarefaction curves analyses, we made as a routinely quality assurance, showed that the sequencing deep was comparable with no more bacteria to be discovered. See the figure below (attached file).

References:

[17] Pérez Lizaur, A.B.; Palacios González B.; Castro Becerra A.L. & Flores Galicia I. Sistema mexicano de alimentos equivalentes. 4ª Ed. México Press; 2014.

--end of text--

Reviewer 2 Report

The authors have addressed almost all my concerns.

However, the paper still shows some inaccuracies and unclear sentences, and minor English edits are still needed.

Please, consider the following:

L77: a statistically. L78: hatheway, and Lachnospiraceae in italics. L81: 272? L84: were the. L96: into L117: Please define ADA at its first occurrence and use the abbreviation later (L120). Exclusion criteria: what about pre/probiotics intake? L145: as overweight …. as obesity. L146-151: please check and rephrase. L198: 16S rRNA. L201: dNTPs. Please specify the Taq used. L219: do the authors mean high-quality reads? And how many OTUs did they obtain? L221: accessed. L259: using PICRUSt. L288: only at phylum level? Table 1: METs L373: in terms. Figure 2 legend: writing only “Observed” makes no sense. You should refer to the “number of observed OTUs” (here, in materials and methods and throughout the manuscript). Please revise the wording of tables (Table or table) and figures (Figure S02). Figure 3 legend needs to be rewritten. Just a suggestion: “Bacterial beta diversity. Principal Coordinates Analysis based on unweighted UniFrac distances between the gut microbiota profiles of individuals from the six groups. Control (CO), red triangles; …” (no need to use the word color!). L420: no capital letter for weight Figure 5 legend: Enterococcaceae and no capital letter for age. Last sentence “indicates de participants”? Please rephrase. L445-448: it makes no sense to report which groups are different in alpha diversity, rather where biodiversity is greater or lesser. L464: please add the unit of measure. L465: Acidobacteria L465-467: this is speculation (and not related to the previous sentence dealing with Bacteroidales). If you want to keep it, please tone down. L477: Comamonadaceae L483: another overstatement (Prevotella is also typically related to high-fiber diets). Please tone down or specify that its role is debated. L491: were. 495-503: the correlations found out by the authors are often conflicting with literature. Please, stress this point. L507: no capital letter for energy, carbohydrate, amino acid and lipid. L512: which groups? It is not clear what the authors are referring to. L541: no capital letter for healthy and prediabetes.

Author Response

2019-12-31 Tuesday

Manuscript ID: microorganisms-661237

Title: Characterization of the gut microbiota of individuals at different T2D stages reveals a complex relationship with the host.

Authors: Alejandra Chávez-Carbajal, María Luisa Pizano-Zárate, Fernando Hernández-Quiroz, Guillermo Federico Ortiz-Luna, Rosa María Morales-Hernández, Amapola De Sales-Millán, María Hernández-Trejo, Angelina García-Vite, Luis Beltrán-Lagunes, Carlos Hoyo-Vadillo, Jaime García-Mena.

Reviewer 2

Comments and Suggestions for Authors

The authors have addressed almost all my concerns. However, the paper still shows some inaccuracies and unclear sentences, and minor English edits are still needed.

Answer: Following the Reviewer 2 suggestion, the text has been reviewed again by a native English speaker.

Please, consider the following:

-L77: a statistically.

Answer: The change was made on L77.

-L78: hatheway, and Lachnospiraceae in italics.

Answer: The change was made on L78.

-L81: 272?

Answer: We thank the reviewer for pointing out this confusing number, the name “Roseburia_272” was making reference the name as reported in green genes v13.8 database. The number has been removed from L81.

-L84: were the.

Answer: The error was corrected in L84.

-L96: into L117: Please define ADA at its first occurrence and use the abbreviation later (L120).

Answer: American Diabetes Association (ADA) is first defined in L116 and abbreviation is used later as requested.

-Exclusion criteria: what about pre/probiotics intake?

Answer: many thanks for the observation, the use of prebiotics or probiotics was an exclusion criterion, the information was added in L127.

-L145: as overweight …. as obesity.

Answer: The change was made on L150.

-L146-151: please check and rephrase.

Answer: The change was made on L150-152. “The waist circumference was measured at the midpoint between the lower rib and iliac crest. The hip circumference was measured under the iliac crest using a metric tape, in all subjects (Lufkin, model W606ME, Lufkin Board, Cleveland, Ohio, United States of America).”

-L198: 16S rRNA.

Answer: The change was made on L213 to rDNA.

-L201: dNTPs. Please specify the Taq used.

Answer: The dNTP Mix (10 mM each), Cat. R0193 Thermo Scientific; and Taq polymerase used Phusion High-Fidelity DNA Polymerase (Cat. F-530L Finnzymes-Thermo Scientific) are now indicated in the text L216-L217.

-L219: do the authors mean high-quality reads? And how many OTUs did they obtain?

Answer: After sequencing, reads were filtered by the PGM software to exclude low quality and polyclonal sequences, obtaining reads with Phred33 Quality Score of 31 on average. Afterward DNA sequences were classified into 33,000 Operational Taxonomic Units (OTUs) using closed based picking parameters with a 97 % similarity level against Greengenes database v13.8. This information was entered in L228 and L231-234.

-L221: accessed.

Answer: The conjugation error was corrected in L436.

-L259: using PICRUSt.

Answer: The requested change was made on L259.

-L288: only at phylum level?

Answer: a correction was made at L366.

-Table 1: METs

Answer: a correction was made in Table 1.

-L373: in terms.

Answer: The change was made on L476.

-Figure 2 legend: writing only “Observed” makes no sense. You should refer to the “number of observed OTUs” (here, in materials and methods and throughout the manuscript).

Answer: In our analysis of alfa diversity, “Observed” refers to the total of bacteria observed, the alpha analyses output calls this form. However, for clarity, we entered the reviewer’s suggestion in the legend of Figure 2, L572.

-Please revise the wording of tables (Table or table) and figures (Figure S02).

Answer: revised and corrected in L490.

-Figure 3 legend needs to be rewritten. Just a suggestion: “Bacterial beta diversity. Principal Coordinates Analysis based on unweighted UniFrac distances between the gut microbiota profiles of individuals from the six groups. Control (CO), red triangles; …” (no need to use the word color!).

Answer: legend was edited following the reviewer’s advice and changed to “Figure 3. Bacterial beta diversity. Principal Coordinates Analysis based on unweighted UniFrac distances between the gut microbiota profiles of individuals from the six groups. Control (CO) red triangles, Prediabetes (PRE) green squares, T2D no medicated (T2D−No−M) dark blue triangles, T2D with Metformin (T2D−M) purple triangles, T2D with polypharmacy (T2D−P) light blue triangles, and T2D with polypharmacy + insulin (T2D−P+I) brown circles (see Table S4)”.

-L420: no capital letter for weight.

Answer: The change was made on L545.

-Figure 5 legend: Enterococcaceae and no capital letter for age. Last sentence “indicates de participants”? Please rephrase.

Answer: The change was made on age. The lasts sentence indicates the number of participants by a group for the D and H parts of Figure 5. The new last sentence says, “numbers in parenthesis indicate the number of participants by group (D and H) (see Table S6).”

-L445-448: it makes no sense to report which groups are different in alpha diversity, rather where biodiversity is greater or lesser.

Answer: we rephrased the whole paragraph as requested, and now it reads “We also found that the alpha- and beta-diversities of the following groups showed significant statistical differences. For instance for alpha diversity (number of observed OTUs), diversity is lower in T2D−M compared to CO, T2D−P+I, or PRE; the diversity based on Chao1 index is lower in T2D−M compared to T2D−P, CO, T2D−P+I, and PRE; the diversity in T2D−P is larger compared to T2D−No−M; the diversity in T2D−No−M is lower compared to CO, T2D−P+I, and PRE; and the diversity in T2D−M based on the Shannon index, is lower compared to CO, T2D−P+I and PRE (Table S3). For the beta diversity, T2D−No−M is lower compared to CO, T2D−P, and T2D−P+I; the beta diversity in T2D−M is lower compared to CO, T2D−P+I and PRE (Table S4)”.

-L464: please add the unit of measure.

Answer: Units mg/dL was made on L590.

-L465: Acidobacteria

Answer: We acknowledge the expertise of the Reviewer 2 on this subject; however, we make reference to the class Acidobacteriia, no to the phylum Acidobacteria. This spelling was obtained from the analysis of similarity against Greengenes database v13.8. The reference [30] was updated to: Dedysh SN, Yilmaz P. Refining the taxonomic structure of the phylum Acidobacteria. Int J Syst Evol Microbiol. 2018 Dec;68(12):3796-3806. doi: 10.1099/ijsem.0.003062. Epub 2018 Oct 16. L591.

-L465-467: this is speculation (and not related to the previous sentence dealing with Bacteroidales). If you want to keep it, please tone down.

Answer: We understand the Reviewer 2 refers to the statement “We observed an increase in the abundance of Pelomonas spp. in T2D−M but not in T2D−No−M group. This finding suggests the increase of this bacterium might be due to Metformin treatment rather than T2D per se.”. This sentence was tone down as follows: “On the other hand, we observed an increase in the abundance of Pelomonas spp. in T2D−M but not in T2D−No−M group. A possible interpretation of this finding is that the increase of this bacterium might be due to Metformin treatment rather than T2D per se” L594-596.

-L477: Comamonadaceae L483: another overstatement (Prevotella is also typically related to high-fiber diets). Please tone down or specify that its role is debated.

Answer: wee added information in the text with the intention of conciliating the opportune observation of the Reviewer 2, L604-L605.

-L491: were.

Answer: The change was made on L722.

-495-503: the correlations found out by the authors are often conflicting with literature. Please, stress this point.

Answer: we added the statement “In our study, we found some interesting conflicting results in relation to published literature” at the beginning of the paragraph. L726.

-L507: no capital letter for energy, carbohydrate, amino acid and lipid.

Answer: The change was made on L739.

-L512: which groups? It is not clear what the authors are referring to.

Answer: we added the name of the groups “PRE and T2D−M” L744.

-L541: no capital letter for healthy and prediabetes.

Answer: The change was made on L856.

--end of text--

Round 3

Reviewer 1 Report

Dear authors,  

I must congratulate you on your willingness in addressing my comments. In my opinion, the clarity and overall quality of the manuscript have been improved. Nonetheless, I still think that the manuscript suffers from several weaknesses such as the study design, which led to non-homogeneous cohort groups, and the data analysis.